# Performance Characterization and Tuning of a Charge-Splitting High Dynamic Range 4-Tap CMOS Image Sensor [note 1]

**DOI:** 10.3390/s25226953

**Published:** 2025-11-13

**Authors:** Yu Feng, Keiichiro Kagawa, Kamel Mars, Keita Yasutomi, Shoji Kawahito

**Affiliations:** 1Graduate School of Science and Technology, Shizuoka University, Hamamatsu 432-8011, Japan; feng.yu.16@shizuoka.ac.jp; 2Research Institute of Electronics, Shizuoka University, Hamamatsu 432-8011, Japan; 3Faculty of Science and Technology, Shizuoka Institute of Science and Technology, Fukuroi 437-8555, Japan

**Keywords:** complementary metal-oxide-semiconductor (CMOS) image sensor (CIS), high dynamic range (HDR), multi-tap pixel, charge-splitting, motion artifact and light emitting diode (LED) flicker mitigation

## Abstract

Single-exposure high dynamic range (HDR) imaging is critical for applications such as automotive and surveillance cameras, where motion artifacts and light emitting diode (LED) flicker are significant challenges. Charge-splitting HDR imaging using multi-tap complementary metal-oxide-semiconductor (CMOS) image sensors (CIS) effectively mitigates these issues and offers programmable dynamic range extension, demonstrating significant potential for such applications. In this work, we present a model to describe the performance of the charge-splitting pixel. Then, we experimentally characterize and tune the performance of a 4-tap CIS. Through performance tuning, the image sensor achieves a single-exposure dynamic range (DR) of 126 dB. This represents an improvement of 16 dB over the previously reported 110 dB while maintaining a high signal-to-noise ratio (SNR), with a minimum transition SNR exceeding 30 dB.

## 1. Introduction

The demand for high dynamic range (HDR) imaging has grown significantly across fields such as surveillance and automotive cameras. This growth is driven by the need to capture high contrast scenes without sacrificing the detail in either brightly illuminated or shadowed areas. The conventional multi-exposure HDR (MEHDR) method achieves a programmable dynamic range (DR) extension by capturing a scene with different exposure times and combining the frames into a single HDR image [1]. However, MEHDR is susceptible to motion artifacts and light emitting diode (LED) flicker, which makes it unsuitable for dynamic scenes [2]. In such scenarios, single-exposure HDR becomes a more critical requirement. Automotive applications such as advanced driver assistance systems (ADAS) or autonomous driving require cameras to operate reliably under dynamic and extreme outdoor lighting conditions. In these cases, a single-exposure HDR of over 120 dB is often required to ensure safety and functionality [3,4].

Significant progress has been made in extending the dynamic range (DR) of complementary metal-oxide-semiconductor (CMOS) image sensors (CIS) to meet these demands. HDR CISs now achieve single-exposure HDRs over 100 dB, compared to the 60–70 dB for standard DR CISs. Notable technologies include dual conversion gain (DCG), split-photodiodes (split-PDs), and lateral overflow integration capacitor (LOFICs), which are often used in combination. DCG extends DR by switching between high and low conversion gains during readout [5]. The split-PD method uses two photodiodes of different sizes and photosensitivities per pixel to capture a scene [6]. LOFIC achieves HDR by accumulating excess charge in in-pixel overflow capacitors [7,8]. While these methods, individually or in combination, offer high single-exposure DRs and can mitigate motion artifacts and LED flicker, they have inherent drawbacks. LOFIC requires a large in-pixel capacitor, which reduces the fill factor or requires a modification to the fabrication process. Furthermore, the DR of these methods is determined by the pixel design and is fixed during fabrication. Therefore, none of these approaches offer programmable DR, which makes them difficult to adapt to various applications.

To address these limitations, we previously introduced a programmable HDR imaging scheme based on a prototype 4-tap CIS [9]. We applied the charge-splitting HDR method [10] to the 4-tap CIS, in which the sensor splits and stores the photogenerated charge among the four taps according to programmed exposure duty cycles. This process produces four images with different photosensitivities, enabling the sensor to dynamically extend the DR and provide flexibility for adjustments based on the scene or application. Although this prototype 4-tap CIS was originally designed for the ultra-fast charge transfer required in short-pulse time-of-flight (ToF) depth imaging applications [11], its 4-tap architecture enables us to apply the charge-splitting HDR method. Its fast charge transfer speed allows for charge-splitting at a speed much higher than the imaging frame rate. This effectively mitigates motion artifacts and LED flicker. Our previous work demonstrated that this system could achieve a single-exposure HDR of up to 110 dB while maintaining high SNR across the range [12].

This paper describes the principles determining the dynamic range of the charge-splitting HDR method and presents a comprehensive experimental characterization followed by performance tuning of the 4-tap CIS-based charge-splitting HDR system. We explore the critical trade-offs between DR, SNR, and linearity. We demonstrate that the tuned HDR system achieves a single-exposure HDR of 126 dB while maintaining a minimum transition SNR above 30 dB.

## 2. Sensor Architecture and HDR Operation

### 2.1. A 4-Tap CMOS Image Sensor

Our programmable HDR system is based on a 4-tap CIS architecture, which is fundamental to our charge-splitting HDR method. The prototype 4-tap CIS was developed in our laboratory based on the lateral electric field charge modulator (LEFM) technology [13]. LEFM enables ultra-fast charge transfer at the nanosecond scale and supports scalable multi-tap architectures. The 4-tap CIS is designed for ToF applications, where it operates lag free when the time window is set at 11 ns [11]. Figure 1a illustrates the pixel’s structure, and Figure 1b shows the equivalent pixel circuit. The 4-tap pixel comprises a single photodiode (PD), shared among four individual taps and a drain. Each tap consists of a transfer gate (G) and a floating diffusion (FD), in a three-transistor (3T) configuration, optimized for ultra-fast charge transfer. This contrasts with a conventional CIS pixel, which typically has only one FD.

The unique structure of the 4-tap pixel enables its four FDs to function as four independent memory nodes. The FDs are reset via a reset signal before exposure. During exposure, a potential gradient within the pixel guides the photogenerated charges from the PD, through a charge corridor, to the front of the transfer gates. The charge corridor is the shared charge transfer path from the photodiode to the gates, indicated by red dashed lines in Figure 1. Then, one transfer gate is turned on at a time, connecting the PD to a specific FD for charge transfer. These charges are transferred to and stored in the corresponding FD. The full well capacity (FWC) of each tap is determined by the capacity of its FD. Additionally, the drain gate (GD) can be turned on to drain the photogenerated charge. This creates an insensitive period during the image readout. Unlike conventional CIS pixels, which output a single value per pixel after readout, the 4-tap pixel provides four distinct pixel values, one from each of its FDs. This is possible because the four transfer gates operate independently, capturing four pixel values that correspond to four different exposure patterns within a single exposure period (or one image frame).

A micrograph of the prototype 4-tap CIS is shown in Figure 2. Table 1 summarizes the key characteristics of the CIS. The sensor was fabricated using a 0.11 µm CIS process and has a relatively large pixel size of 16.8 µm × 16.8 µm. Originally designed for ToF applications, the pixel has a high FWC of 40 k e^−^ and a relatively low conversion gain of 22.2 µV/e^−^. These parameters were chosen to ensure high depth precision in ToF depth measurement.

### 2.2. Principle of High Dynamic Range Imaging via the Charge-Splitting Method

The DR of a CIS is defined as the ratio between the maximum number of electrons ($N_{sat}$) at the sensor’s FWC (i.e., saturation) and the equivalent number of electrons of the dark noise floor ($N_{noise}$) when $N_{noise}$ is relatively high [4]. This relationship is typically expressed in decibels (dB):(1)$$DR=20\log_{10}\left(\frac{N_{sat}}{N_{noise}}\right).$$

The conventional MEHDR method extends DR by capturing the same scene using different exposure times, with long exposures for dark regions and short exposures for bright regions. An HDR image is then synthesized from these differently exposed frames. The resulting DR extension is determined by the ratio of the longer and shorter exposure times, as denoted by the following equation:(2)$${DR}_{ext}=20\log_{10}\left(\frac{T_{long}}{T_{short}}\right).$$

The conventional MEHDR method can be emulated using the 4-tap CIS. This is achieved by assigning four distinct exposure times to each of the four taps, enabling the sensor to capture multiple images with different photosensitivities in a single acquisition cycle. Figure 3a illustrates the process to capture four frames of image with four different photosensitivities. During the exposure period, a high gate voltage indicates that the corresponding gate is “ON,” allowing photoelectrons to accumulate in the designated tap. Conversely, a low gate voltage indicates the gate is “OFF”. As such, the effective exposure time for each tap is determined by the gate’s duty cycle. Furthermore, the duty cycle of the gates determines the DR of the synthesized HDR image, according to Equation (2). Each tap of the sensor is in a three-transistor pixel configuration that operates in differential delta sampling mode. During readout, each tap’s FD reset level (RES) is read after the signal level (SIG).

In our 4-tap HDR system, the charge-splitting HDR method operates under the same principle as conventional MEHDR. However, switching occurs more frequently. Figure 3b illustrates the gating timing charts for the charge-splitting method. This method utilizes a 4-tap CIS to split photogenerated charge among its four taps within a single frame’s exposure period, enabling highly flexible and complex exposure patterns limited only by the sensor’s charge transfer speed. To implement the charge-splitting HDR method using the 4-tap CIS, the total exposure time for each tap is divided into numerous shorter cycles ($\tau_{c}$), which are repeated N times throughout the frame period. This repetition ensures that the effective exposure time for each tap, defined by the gate duty cycle and total exposure period, remains consistent with that of the conventional MEHDR method. Dividing a single, long exposure into numerous shorter cycles offers key advantages, particularly for HDR imaging of dynamic scenes. Within each short cycle, the timing difference among the images captured by different taps is shortened, significantly reducing motion artifacts. Effectively, a large number of near-simultaneous snapshots of the subject are captured and accumulated in each tap over the total exposure period, mitigating motion artifacts typically observed in conventional methods. In contrast, in the conventional MEHDR method, which captures four images with long, distinct exposure times sequentially, there is an inevitable introduction of more prominent motion artifacts. The ability to mitigate motion artifacts improves as the cycle length $\tau_{c}$ is shortened and the number of cycles increases, and a shorter minimum exposure time at the least-photosensitive tap enhances the achievable DR extension. The minimum gate duration within each short cycle for this sensor is limited by its shortest time window of 20 ns, as mentioned in Table 1.

Table 2 shows an example of operation parameters for the charge-splitting HDR method. It details the photosensitivity settings, total exposure time, duty cycle, expected DR extension, and the cycle exposure time τ within the total exposure time for the charge-splitting HDR method at each of the four taps.

The combined total exposure time for the 4-tap sensor for both methods is 31.11 ms, and the readout time for the sensor is 1.45 ms, as mentioned in Table 1. The number of repeated exposure cycles N is 100. The frame interval for the synthesized HDR image is 32.56 ms, equivalent to a typical video frame rate of 30.7 fps.

The total DR ($DR_{total}$) of the charge-splitting method is determined by the base DR of the CIS ($DR_{base}$) and the extended DR ($DR_{ext}$) achieved through combining different frames, as described by the following:(3)$$DR_{total}=DR_{base}+\;DR_{ext}.$$

The extended DR for a 4-tap CIS comes from combining the longest exposure image from Tap 1 with three images with shorter exposure times from the remaining taps. The relationship between the exposure times and the resulting extended DR is derived from Equation (2) and expressed as follows:(4)$$DR_{ext}=\sum_{n=1}^{3}20\log_{10}\left(t_{n}/t_{n+1}\right),$$
where $t_{n}$ is the effective exposure time for tap n. For our configuration, the exposure time ratio for two adjacent taps is 10×. Based on these settings, a DR extension of 20 dB is expected by combining two adjacent taps, and the ideal extended dynamic range $DR_{ext}$ for the 4-tap CIS is calculated to be 60 dB.

## 3. Characterization of Dynamic Range and Operating Conditions

This section presents a model describing the key factors that determine the sensor’s HDR performance, focusing on the trade-offs between FWC, crosstalk, and key operating voltages.

### Factors Determining Sensor Performance

In the proposed 4-tap HDR system, DR is a key performance metric. As described in Equation (1), the sensor’s base DR is fundamentally determined by the noise floor and the FWC of the FD. While the noise floor is difficult to adjust, the FWC can be tuned by adjusting the operating voltages of reset transistor and charge transfer gates.

Figure 4 shows the potential diagram of the charge modulator for Tap 1 and Tap 4 as a representative example. The FWC, which defines the upper limit of charge storage, is typically set by the low voltage of the reset transistor’s gate ($V_{RES-L}$). When a tap is approaching saturation, its reset transistor acts as an anti-blooming drain, allowing excessive charges to be drained through the reset transistor. The voltage $V_{RES-L}$ thus determines the height of the reset transistor’s potential barrier. Although a higher potential barrier (i.e., a lower $V_{RES-L}$) would ideally increase the FWC and maximizing the sensor’s base DR, it introduces a critical trade-off with residual charge crosstalk. If the potential height of the charge corridor is not sufficiently higher than that of the reset transistor, excessive charges cannot be drained effectively. Instead, they remain in the charge corridor as residual charges. As more residual charges accumulate, some charges can overcome the transfer gate potential barrier of inactive taps and reach the corresponding FDs. This crosstalk degrades the signal integrity of those taps, which limits the DR extension and linearity. As shown in Figure 4, the potential barriers at the transfer gates are determined by the low voltage of the transfer gate voltage ($V_{G-L}$). Therefore, the parameters $V_{RES-L}$ and $V_{G-L}$ must be carefully tuned to achieve a delicate balance between maximizing FWC for a high DR and maintaining sufficient potential barriers to suppress residual charge induced crosstalk.

There are two other forms of crosstalk in the pixel: optical and diffusion-current-based crosstalk. Figure 5 illustrates a cross-sectional view along the A-A’ line of the pixel, which contains three n-type layers with varying doping concentrations to create the desired potential gradients. The deeper n2 regions assist in collecting photoelectrons in the deep epi layer into the photodiode. A p-n junction between the epi-layer and the FD, indicated by the red dotted lines in Figure 5b, is designed to prevent diffusion current from entering the FD. However, under high illumination, incident light can reflect off the gate, bypass the metal light shield, and directly reach the FD, causing optical crosstalk. Moreover, a large diffusion current can cause charges to overcome the p-n junction barrier and accumulate in an inactive FD. These two phenomena compromise signal integrity and reduce linearity, particularly in the low-photosensitive taps (i.e., Taps 3 and 4). This performance degradation, caused by optical crosstalk and diffusion current, is an inherent limitation of the pixel design that constrains the maximum DR and linearity.

## 4. Sensor Measurements and Performance Tuning

Building on the factors that determine HDR performance, which were discussed in Section 3, this section details the experimental tuning of the sensor under varying operating parameters. Our tuning process begins with characterizing the potential profile of the pixel. Next, we establish performance baselines and identify the optimal tap configuration. Finally, we fine-tune the key operating voltages to maximize dynamic range extension.

### 4.1. Potential Measurements Through Charge Injection

As a first step, we performed a charge injection experiment to measure the potential height of the charge corridor [14,15]. This potential is a critical parameter, as discussed in Section 3, that governs the trade-off between FWC and residual charge crosstalk. Figure 6 shows the pixel layout and the potential at the cross-section view along the A-B line and A- A’ line of the pixel. In this experiment, the drain gate was always ON. By decreasing the drain voltage ($V_{DR}$), charges were injected from the drain to the charge corridor. Simultaneously, the four transfer gates were switched ON during charge injection, allowing the charges to be transferred to all four taps. During readout, the charges in each of the four FDs were read. The injection time was set to 31.11 ms, being consistent with the HDR measurements. The drain voltage $V_{DR}$ was swept from 1 V to 2.7 V in 0.05 V steps. Figure 7 shows the timing chart for the charge injection experiment. $V_{FD1}$-$V_{FD4}$ represent the example voltage at each of the four FDs. Turning on G1-G4 starts the accumulation of the injected currents in the FDs. After the injection time, G1-G4 are turned off and $V_{FD1}$-$V_{FD4}$ are read out. Finally, the floating diffusions are reset.

Figure 8 shows the measured normalized pixel values in the charge injection measurement at each tap with varying $V_{RES-L}$. The pinning voltage is calculated by the point at which the linearly fitted line of the pixel values intersects the horizontal axis, with $V_{DR}$ ranging from 2.0 V to 2.2 V. As an example, the linearly fitted pixel values of Tap 4 are shown as dashed line. This intersection represents the point at which a large amount of charges start flowing into the FDs, which is the pinning point of the charge corridor by the injected charge. Table 3 summarizes the measured pinning voltages of the four taps. Although a slight increase in pinning voltage was observed as the reset low voltage increased, the pinning voltage remained at approximately 2.2 V across all taps. These measured pinning voltages represent the potential height at the charge corridor. As discussed in Section 3, this potential, along with $V_{RES-L}$, affects the residual charge in the charge corridor, ultimately affecting the DR and linearity of the system. The pinning voltage of the charge corridor is exclusively determined by the p-n junctions within the charge corridor itself. As shown in Table 3, the measured potential height of the charge corridor exhibits minimal correlation with $V_{RES-L}$. We also observed that the pinning voltage measured by Taps 4 is slightly higher than those of Taps 1–3. This result will be discussed further in Section 4.2.1.

### 4.2. HDR Performance Tuning

Next, we measured the photo response of the sensor at varying settings using a luminance box (LB-8623, Kyoritsu Electric, Shizuoka City, Japan) to determine the optimal parameters for achieving high extended DR and high linearity. The luminance level ranged from 0.1492 to 18,243 cd/m^2^. The exposure time was 31.11 ms, and a 50 mm/F2.0 VIS-NIR lens (Edmund Optics, Tokyo, Japan) was used.

#### 4.2.1. Draining Performance

To establish a technical performance ceiling, we first measured the sensor’s draining performance. This experiment quantifies the maximum amount of crosstalk that occurs when all transfer gates are OFF, representing the technical limit for crosstalk reduction. The drain gate was always ON during both exposure and readout, as shown in Figure 9a. The resulting pixel values represent the charges accumulated in the FD, either by overflowing the gate barrier or through optical crosstalk and diffusion current. Figure 9b shows the measured photo response of the four taps, with the x-axis representing the luminance and the y-axis denoting the pixel values. Table 4 summarizes the measured maximum light intensity at saturation (*E_MAX_*) and gamma of the four taps. The gamma (γ) represents the linearity of the photo response and is defined by the following equation:Pixel value = A × (Light intensity)^γ^.(5)

Here, A is a constant value. The mere presence of a signal in the taps indicates that the drain cannot remove all excess charges under high illumination. This leads to crosstalk and eventual saturation. The measurements also reveal that the taps have different saturation values due to differences in crosstalk susceptibility; Tap 3 has the least crosstalk, while Taps 1 and 2 saturate more easily. The draining results can be attributed to the proximity of Taps 1 and 2 to the photodiode in the pixel design, making it easier for excess charges or stray light to reach them through crosstalk. Conversely, Taps 3 and 4 are further away from the PD, making them harder to reach. The photo response and saturation of the taps serve as the practical performance ceiling, defining the technical minimum photosensitivity and maximum DR achievable by the sensor. However, it is important to note that in charge-splitting HDR, the drain gate is typically OFF during exposure. Therefore, this technical minimum photosensitivity defines the ultimate performance and cannot be achieved during imaging.

The measured results of the potential height of the charge corridor in Figure 8 and the draining performance in Figure 9 indicate that the four taps have different gate potential barrier heights, as illustrated in Figure 10. Taps 1 and 2 saturate before Taps 3 and 4, due to their proximity to the photodiode, as shown in Figure 9b. Furthermore, while in the pixel design, the four taps ideally should have the same gate potential barrier heights; horizontal mask misalignment can cause the difference in barrier height between G1 and G2 and between G3 and G4.

#### 4.2.2. Gating Performance

Having established the baseline, we then evaluated the gating performance of each individual tap. The goal of this experiment was to identify crosstalk among the taps. The drain gate was OFF during exposure and ON during readout. For each measurement, the transfer gate for one tap was switched ON, while the others were OFF. Figure 11 shows the measured photo response of the four taps. Table 5 compares the light intensity needed for each tap to reach 1000 LSB. The results are consistent with the previous experiment in Section 4.2.1. Taps 3 and 4 generally require a higher light intensity to saturate, while Taps 1 and 2 saturate more easily. We observed the same phenomenon as before, in that Tap 3 exhibits the least crosstalk, as it takes the highest incident illuminance to saturate in all settings (except when Tap 3 itself is active).

One outlier was observed in Figure 11a. When Tap 1 is active, the inactive Tap 4 exhibits greater photosensitivity than Tap 2. This is likely due to the proximity of Taps 1 and 4, where the gating voltage at Tap 1 may have affected the gate barrier at Tap 4 due to proximity coupling. Combining the results of Section 4.2.1 and Section 4.2.2, we find that Tap 3 has the least crosstalk and therefore is suitable for the lowest photosensitivity and highest DR extension. Therefore, in the subsequent HDR measurements, we assigned Tap 3 as the least-photosensitive tap, with the tap order configured as 1, 2, 4, 3.

#### 4.2.3. HDR Performance with Varying Reset Gate Low Voltage

With the optimal tap configuration established, we proceeded to tune the key operating voltages, beginning with the reset gate low voltage $V_{RES-L}$ by photo response measurements. As described in Section 3, this voltage directly controls the draining of excess charge and is therefore critical for mitigating residual charge crosstalk in the lowest-sensitivity taps. Figure 12a shows the timing chart of the charge-splitting HDR imaging. Based on the findings in Section 4.2.2, for the subsequent HDR measurements, the tap assignments were reconfigured, as shown in Figure 12a, with Tap 3 now designated as the least photosensitive tap to maximize DR. Figure 12b shows the measured photo response of the four taps with varying $V_{RES-L}$. Table 6 summarizes the measured DR and gamma of the four taps.

The photo response results show that although the first to third taps exhibit almost no difference in performance with varying $V_{RES-L}$, the fourth tap’s performance is significantly different, improving as $V_{RES-L}$ increases from 0.5 V to 1.7 V. The dynamic range improved from 18.7 dB to 19.9 dB, with much better linearity, as gamma improved from 0.65 to 0.87. This shows that a higher $V_{RES-L}$ effectively drains excess charge during high light intensity, resulting in fewer residual charges in the charge corridor, thereby improving DR and linearity at the fourth tap, as discussed in Section 3.

However, the measured potential at the charge corridor is approximately 2.2 V (Section 4.1), suggesting that an even higher $V_{RES-L}$ would be needed to prevent residual charges. We conducted measurements with higher $V_{RES-L}$ (1.8 V and 1.9 V), but found that the FWC began decreasing, which would negatively impact HDR performance. Furthermore, at these high voltages, a large portion of the pixels appeared dark, as those pixels were being reset during exposure. This indicates that a $V_{RES-L}$ higher than 1.7 V is not viable due to FWC loss and unintended resets. Therefore, in this work, 1.7 V is the optimal voltage for $V_{RES-L}$.

#### 4.2.4. Transfer Gate Low Voltage vs. Leakage to Taps at High Illumination

Next, we optimized the transfer gate low voltage, $V_{G-L}$. This voltage sets the potential barrier for inactive taps, and tuning it is essential for preventing charge leakage at high illumination. The HDR performance for different $V_{G-L}$ settings are shown in Figure 13. This result shows that although the first and second taps show similar performance, the third and fourth taps show much worse DR extension and linearity with higher $V_{G-L}$. The transfer gate low voltage determines the potential barrier height of each gate when inactive. A higher $V_{G-L}$ results in a lower barrier, which allows more charges to leak to the low-photosensitive taps (i.e., Taps 3 and 4), thereby introducing crosstalk.

Although a higher potential barrier is better, we found that further lowering $V_{G-L}$ below −1.8 V did not improve in DR or linearity. Since the sensor was designed for ToF depth imaging, which requires ultra-fast charge transfer, the achievable gate barrier height is limited by design constraints. Therefore, −1.8 V is the optimal voltage for $V_{G-L}$ for HDR imaging in this work.

#### 4.2.5. HDR Performance with the Shortest Time Window

In the final tuning step, after optimizing the operating voltages $V_{RES-L}$ and $V_{G-L}$, we further improved the system’s performance by minimizing the exposure time of the least-photosensitive tap. Table 7 shows the previous exposure setting and the new setting with the shortest exposure time for the fourth tap. The cycle exposure time for the fourth tap, $\tau_{4}$, was shortened from 280 ns to the shortest achievable time window of 20 ns. The expected DR extension at the fourth tap thus increased from 20 dB to 42.9 dB. Due to the significantly decreased photosensitivity at Tap 4 with the shorter exposure time, the light source’s dynamic range was insufficient to characterize all taps in a single measurement. Therefore, the photo response at the lowest-sensitivity tap (fourth tap, $\tau_{4}\;=\;20\;ns$) was measured separately with the number of exposure cycles, N, increasing from 100 to 800.

Figure 14 shows the measured photo response and the calculated SNR of the four taps with different $\tau_{4}$ settings. Table 8 shows the measured DR and gamma. The SNR is calculated from the following equation [16]:(6)$$SNR=\;20\log_{10}\left(\frac{N_{sig}}{\sqrt{N_{sig}+n_{noise}^{2}}}\right)$$
where $N_{sig}$ is the equivalent number of charges for the signal in the unit of electrons, while $n_{noise}$ is the equivalent number of charges for the dark noise floor. Table 8 shows the total exposure time, cycle exposure time, and the expected DR. The expected DR refers to the theoretically expected DR extension per tap, which is 20 dB or 42.9 dB, derived from Equation (4) for the chosen exposure ratio between adjacent taps.

In Figure 14b, the results show that, due to the tuned HDR performance, with the fourth tap (Tap 3) utilizing the shortest time window $\tau_{4}\;=\;20\;ns$, an HDR of 126 dB was achieved, with a gamma of 0.71 at the fourth tap. For comparison, an HDR of 113 dB was achieved with the previous exposure setting, maintaining a gamma of 0.87 at the fourth tap. For both settings, the minimum transition SNR across the DR exceeds 30 dB, ensuring a high SNR over the entire operational scope. The new setting shows slightly reduced linearity and a lower minimum transition SNR as a trade-off.

In Figure 14a, the result for Tap 3 when drain gate is ON and all taps are OFF during exposure (referred to as “draining”) from Section 4.2.1 is also included for reference. As discussed above, this represents the technical performance ceiling of the pixel and cannot be achieved in practice, as the exposure time cannot be shortened any further and crosstalk occurs during imaging. For both settings, there is also a discrepancy between the expected DR extension and the measured DR extension, which is also due to the crosstalk affecting the fourth tap. As shown in Figure 15, the captured images of an HDR scene were obtained using the 4-tap charge-splitting CIS with the shortest time window at the fourth tap. A high intensity LED (Advanced Illumination, SL244-WHIIC) and a video lens (Pentax, focal length of 12.5 mm, F-number of F2.0) was used. A separator was placed in the middle of the scene. The left side of the scene is dimly illuminated, while the right side features an LED light source. The measured images indicate that the camera effectively captures the dimly illuminated left side as well as the brightly illuminated right side, achieving an HDR of 126 dB. The white dots in the fourth tap image are caused by pixels with a lower gate barrier.

## 5. Discussion

In this work, a single-exposure DR of 126 dB was achieved with the 4-tap CIS-based charge-splitting HDR method through the systematic characterization and tuning of its operating parameters, representing a 16 dB improvement over our previous work. The tuned HDR system offers performance that is competitive with leading single-exposure HDR technologies [17,18,19] required for demanding applications like automotive cameras. Table 9 summarizes the sensor performance comparisons. This 126 dB performance was achieved by systematically mitigating the crosstalk mechanisms detailed in Section 3 through the careful optimization of key operating voltages. Experimental results in Section 4 showed that tuning the reset transistor gate’s low voltage $V_{RES-L}$ was paramount to decreasing the residual charge in the charge corridor. A high $V_{RES-L}$ (optimally 1.7 V) was shown to lower the potential barrier of the reset transistor, which facilitates the draining of excess photogenerated charge under high illumination. Similarly, lowering the transfer gate’s low voltage $V_{G-L}$ (optimally −1.8 V) was necessary to create a sufficiently high potential barrier to prevent charge leakage into inactive taps. However, these optimizations introduced a critical trade-off. For instance, a higher $V_{RES-L}$ would compromise the FWC and could result in dark pixels if the reset transistor activates during exposure.

Despite the successful tuning, the system’s performance was ultimately constrained by the fundamental limitations of the pixel’s architecture, which was originally designed for ToF applications. The measured potential height of the charge corridor (~2.2 V) remained significantly higher than the reset transistor gate’s low voltage $V_{RES-L}$ (optimally 1.7 V), preventing complete elimination of residual charge. Furthermore, the pixel architecture, which prioritizes the ultra-fast charge transfer required for its original ToF applications, does not provide sufficiently high potential barriers at inactive transfer gates to completely prevent charge leakage under intense illumination. The current design also suffers from mismatched FD capacitance and conversion gain, as well as different transfer response at each tap, due to mask misalignment and doping non-uniformity during production. These characteristics necessitate calibration through measurement at least on a per-wafer basis. These factors limit the effectiveness of the tuning and are the primary causes of persistent crosstalk. Consequently, a discrepancy remains between the theoretically expected DR extension at the fourth tap (42.9 dB) and the measured result (32.9 dB) due to the persistent crosstalk mechanisms. While tuning minimized the impact of residual charge, the remaining 10 dB discrepancy between the expected and measured DR extension is likely dominated by the inherent optical and diffusion current-based crosstalk, which is a fundamental limitation of the pixel’s physical architecture rather than its operating voltages. While voltage tuning mitigated residual charge crosstalk, these other forms of crosstalk are limitations of the physical pixel architecture that can be overcome with a redesign. Another limitation is gate performance asymmetry. Tap 3 consistently showed the lowest crosstalk. This has led to the strategic assignment of Tap 3 to the least-photosensitive role, thereby maximizing the achievable DR.

Section 4 also demonstrated the charge-splitting method’s flexibility as a programmable HDR technique with reconfigurable DR/SNR. The final 126 dB DR represents a specific trade-off between maximizing dynamic range and maintaining high linearity (gamma of 0.71 at the fourth tap). Throughout this range, the system maintains high SNR, with a minimum transition SNR exceeding 30 dB. However, for applications where signal fidelity is more critical than absolute DR, the sensor can be reconfigured to deliver a higher minimum transition SNR (>34 dB) at a reduced total DR of 113 dB by setting τ4 to 280 ns.

Future work should focus on a new pixel design specifically optimized for charge-splitting HDR imaging. The design should first focus on improving the base DR of the CIS, which can be achieved by employing a four-transistor (4T) pixel configuration instead of the 3T structure and implementing correlated multiple sampling (CMS) [20] for noise reduction. This would reduce the dark noise floor and result in an improved base DR of more than 70 dB. The design should also aim to minimize optical and diffusion-based crosstalk through improved light shielding and diffusion carrier barrier optimization. Furthermore, the charge corridor potential must be raised to a level higher than that of the reset transistor to ensure the complete draining of excess charge under high illumination. Simultaneously, the potential barriers of inactive transfer gates must be raised significantly to prevent charge from overcoming the barrier and reaching the inactive taps. These improvements will significantly reduce the photosensitivity of the low-sensitivity taps and further extend the DR, approaching the expected DR extension of 80 dB with a 4-tap CIS. This approach would also improve the linearity of the least-photosensitive tap. When added to an improved base DR of 70 dB, a total DR of up to 150 dB can be achieved. This, in turn, will unlock the full potential of multi-tap, charge-splitting sensors as flexible, viable solutions for next-generation HDR imaging systems.

## 6. Conclusions

In this work, we successfully characterized and tuned the HDR performance of a 4-tap CIS HDR system based on the charge-splitting method. By systematically optimizing key operating parameters, we improved the system’s performance to achieve a high single-exposure DR of 126 dB, representing a 16 dB improvement over previously reported results. This was achieved while maintaining a high SNR across the entire range, with a minimum transition SNR exceeding 30 dB. These results confirm that the charge-splitting method is a highly flexible and robust approach for programmable, single-exposure HDR imaging, with strong potential for demanding applications such as automotive and surveillance cameras. Future work on a multi-tap pixel design specifically optimized for charge-splitting HDR could further enhance the HDR performance.

## Figures and Tables

**Figure 1 sensors-25-06953-f001:**
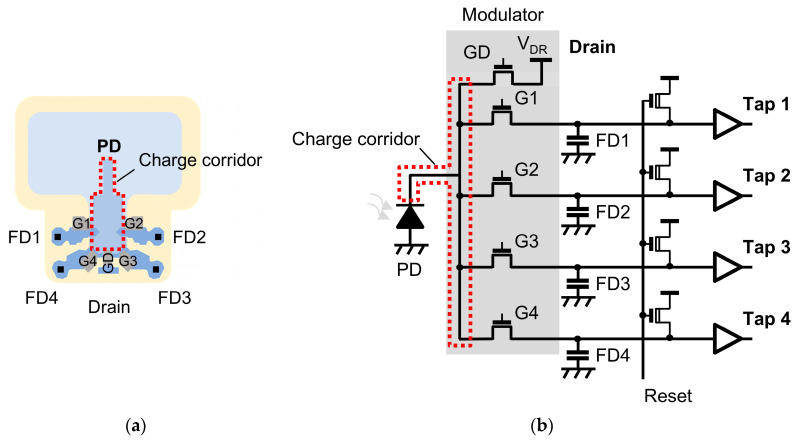
(**a**) The 4-tap pixel diagram and (**b**) the pixel circuit. The charge corridor is indicated by red dashed lines, which is shared by the gates.

**Figure 2 sensors-25-06953-f002:**
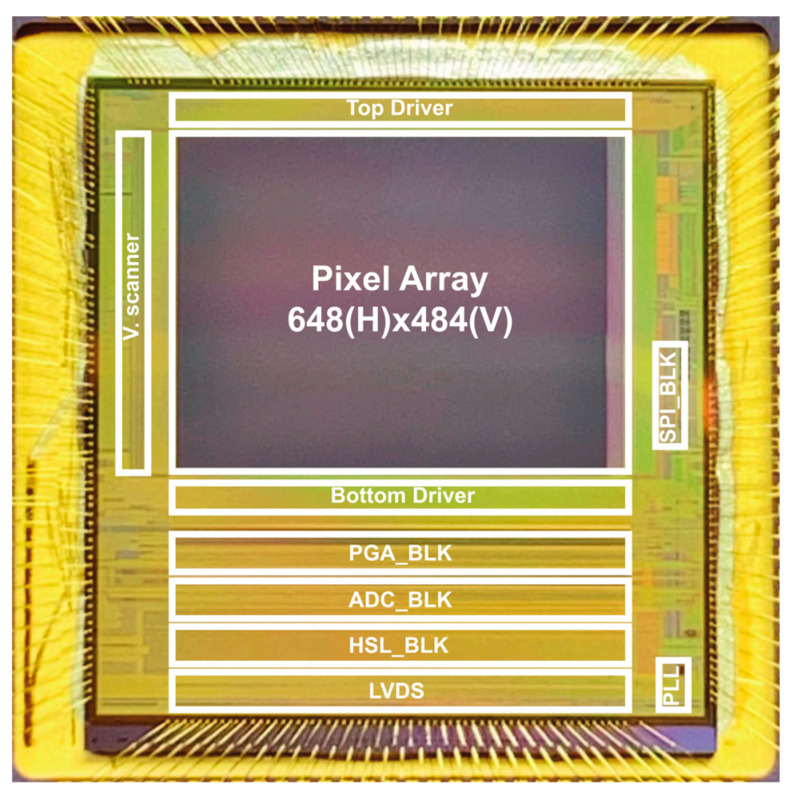
A micrograph of the prototype chip. V. scanner: vertical scanner. SPI_BLK: serial–parallel interface block. PGA_BLK: programmable gain amplifier block. ADC_BLK: analog-to-digital converter block. HSL_BLK: horizontal scanner logic. LVDS: low-voltage differential signaling. PLL: phase-locked loop.

**Figure 3 sensors-25-06953-f003:**
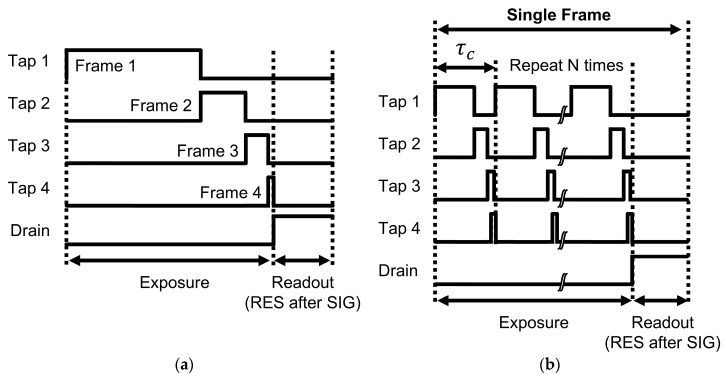
Gating timing charts for (**a**) the conventional multi-exposure HDR method emulated using the 4-tap CIS; (**b**) the charge-splitting HDR method.

**Figure 4 sensors-25-06953-f004:**
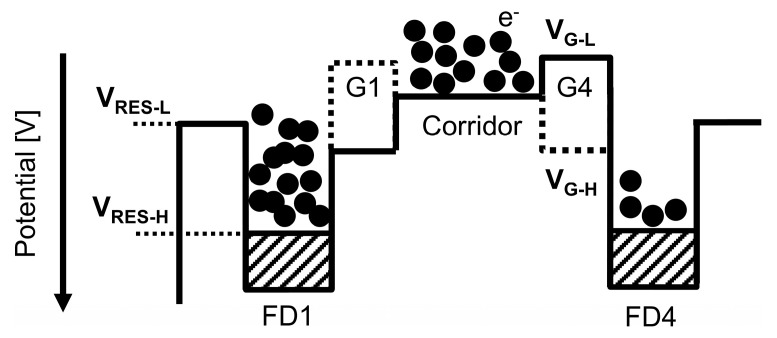
The potential diagram of the charge modulator.

**Figure 5 sensors-25-06953-f005:**
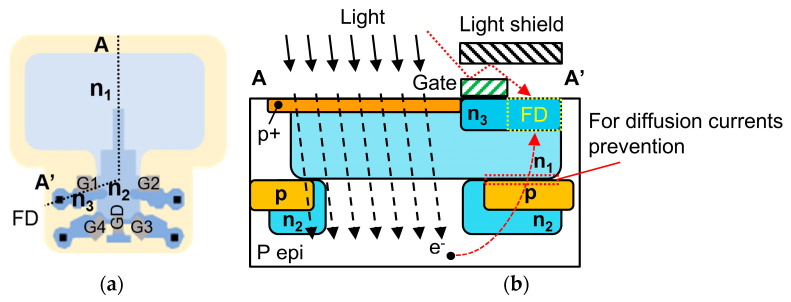
(**a**) The top view of the pixel and (**b**) the cross-section view along the A-A’ line, where optical crosstalk and diffusion current causes charges to accumulate in the inactive tap.

**Figure 6 sensors-25-06953-f006:**
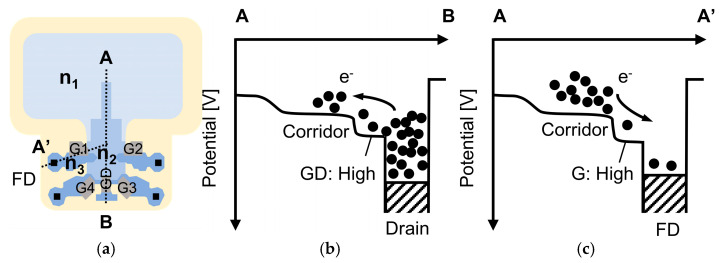
(**a**) The top view of the pixel, (**b**) the cross-section potential along the A-B line, and (**c**) the cross-section potential along the A-A’ line.

**Figure 7 sensors-25-06953-f007:**
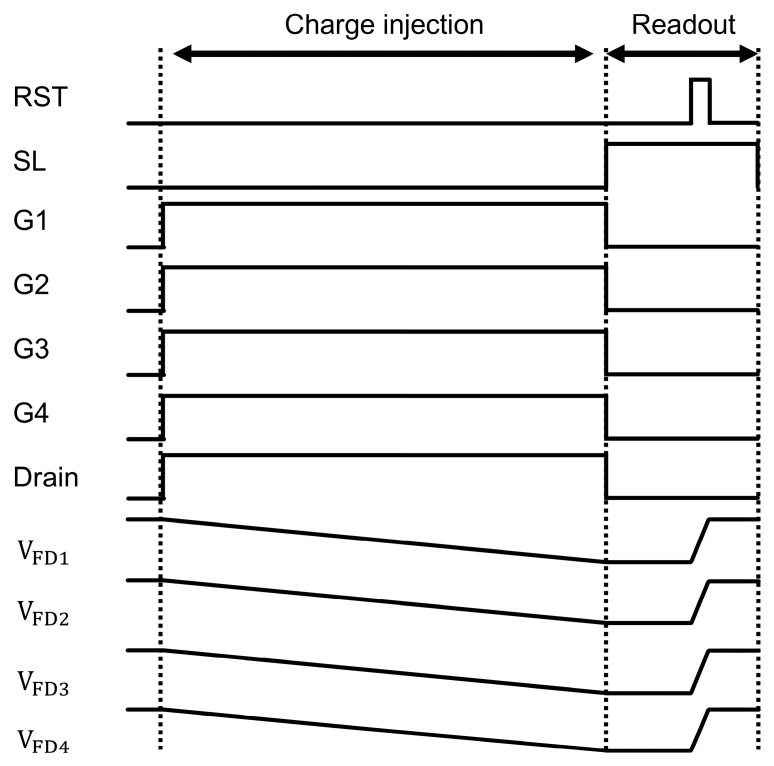
The timing chart for the charge injection experiment.

**Figure 8 sensors-25-06953-f008:**
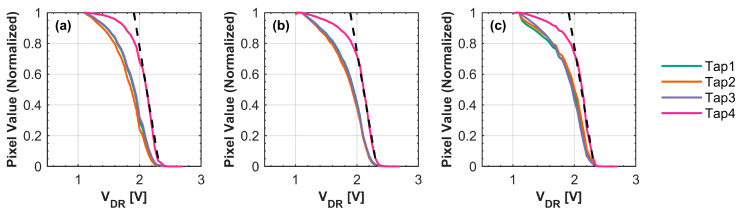
The charge injection experiment results with all gates ON. (**a**) V_RES-L_ = 0.5 V, (**b**) V_RES-L_ = 1.1 V, and (**c**) V_RES-L_ = 1.7 V. The dashed line shows the linearly fitted pixel values at Tap 4 with $V_{DR}$ ranging from 2.0 V to 2.2 V.

**Figure 9 sensors-25-06953-f009:**
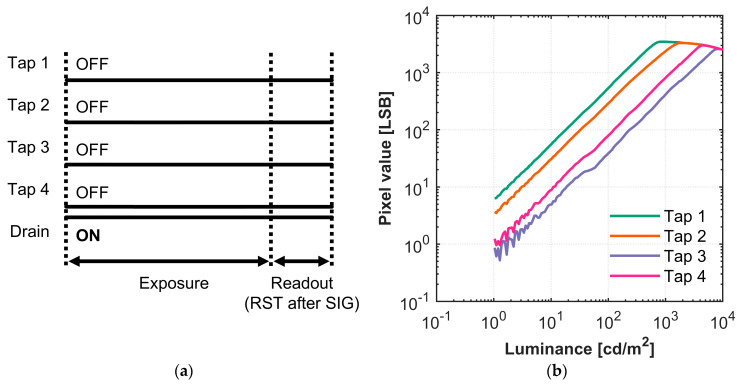
(**a**) The timing chart for the draining measurement while all taps are turned OFF and drain turned ON during exposure; (**b**) luminance versus pixel value curve for the four taps.

**Figure 10 sensors-25-06953-f010:**
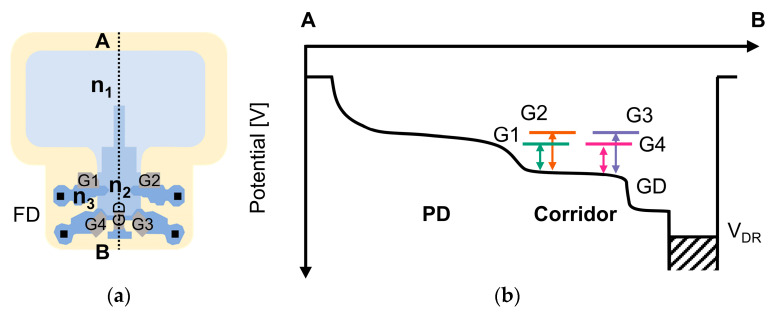
(**a**) The 4-tap pixel diagram and (**b**) the potential barriers difference in the 4 transfer gates when G1-G4 are OFF and GD is ON.

**Figure 11 sensors-25-06953-f011:**
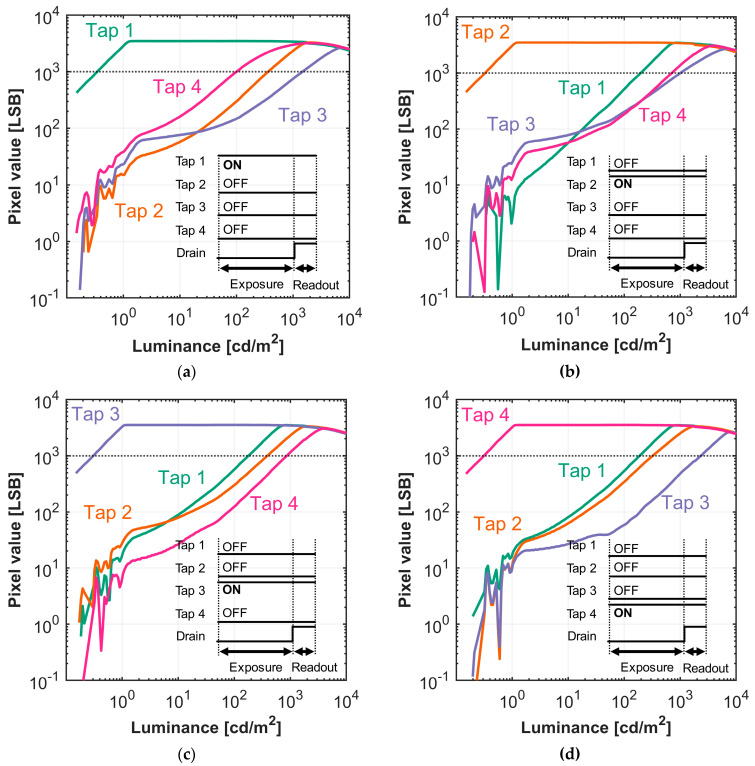
Luminance versus pixel value curve of the 4-tap image sensor while only one of taps is turned ON during accumulation. (**a**) Tap 1 active, (**b**) Tap 2 active, (**c**) Tap 3 active, (**d**) Tap 4 active.

**Figure 12 sensors-25-06953-f012:**
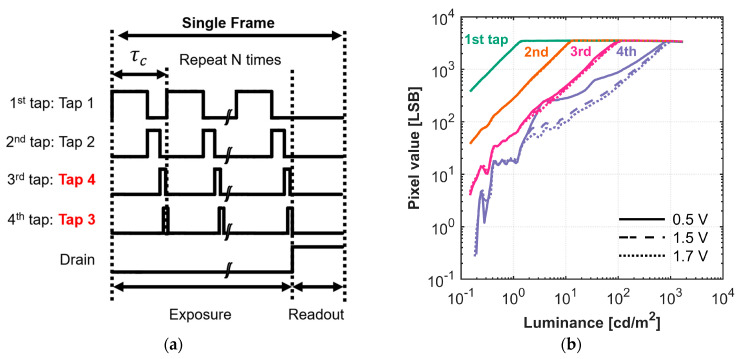
(**a**) The tap setting of the HDR system and (**b**) luminance versus pixel value curve of the 4-tap image sensor with a reset gate low voltage range from 0.5 V to 1.7 V.

**Figure 13 sensors-25-06953-f013:**
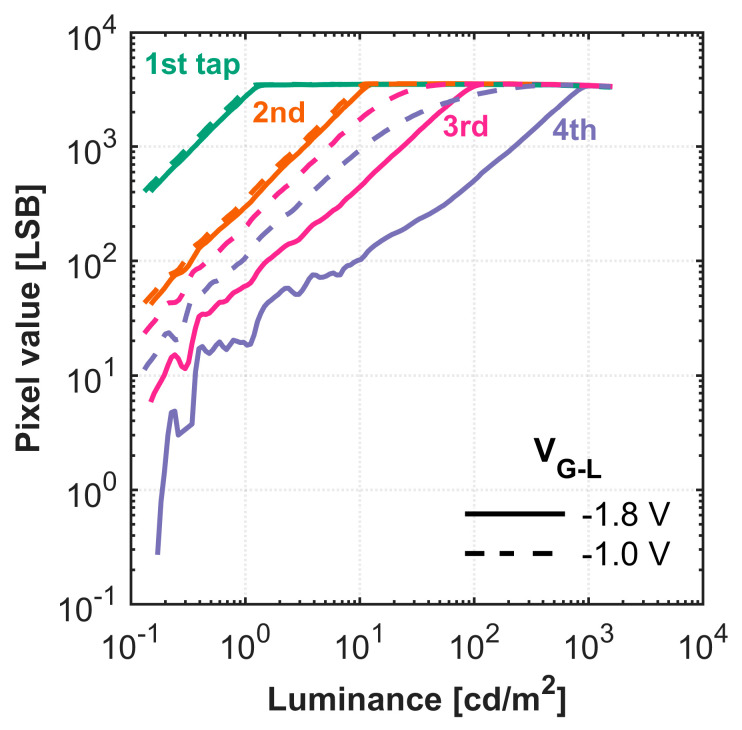
Luminance versus pixel value curve of the 4-tap image sensor with different transfer gate low voltage.

**Figure 14 sensors-25-06953-f014:**
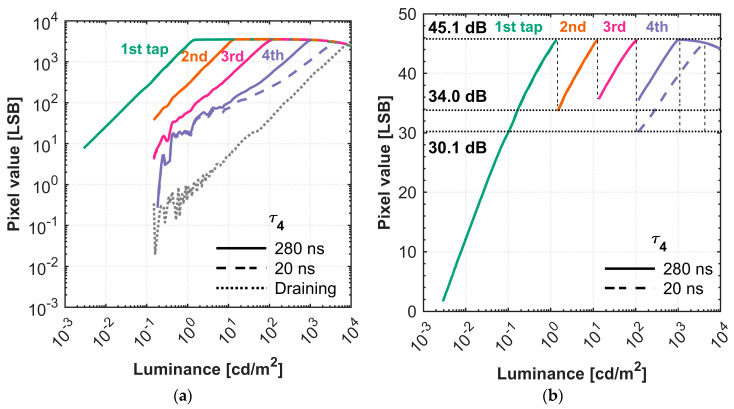
(**a**) Luminance versus pixel value curve and (**b**) the SNR curve of the 4-tap image sensor with shortest time window.

**Figure 15 sensors-25-06953-f015:**
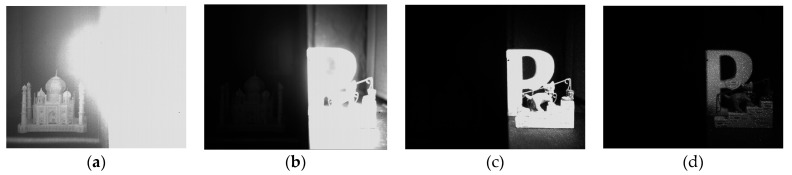
A sample of captured images of an HDR scene from the four taps: (**a**) first tap, (**b**) second tap, (**c**) third tap, (**d**) fourth tap, using the 4-tap image sensor and the shortest time window at the fourth tap.

**Table 1 sensors-25-06953-t001:** Specifications of the prototype 4-tap CIS.

Parameter	Value
Technology	0.11 µm CIS process
Pixel count	648 (H) × 480 (V)
Pixel size	16.8 µm × 16.8 µm
Chip size	14.92 mm × 15.5 mm
Shortest time window	20 ns
ADC resolution	12-bit
Readout time	1.45 ms
Full well capacity *	40 k e^−^
Dark noise **	69 e^−^_RMS_
Dark current	254 e^−^ @ tap 1 (Exposure: 28 ms)
Conversion gain	22.2 µV/e^−^
Quantum efficiency	18.6% @ 940 nm

* Average capacity of the four FDs. ** Dark noise measured at analog gain of 1.

**Table 2 sensors-25-06953-t002:** Example parameters of charge-splitting HDR method.

Tap #	Photosensitivity	Total Exposure Time	Cycle Exposure Time (*τ*)	Duty Cycle	Ex pected DR Extension
1	High	28 ms	280 μs	90%	-
2	Medium	2.8 ms	28 μs	9%	20 dB
3	Low	0.28 ms	2.8 μs	0.9%	20 dB
4	Extremely low	0.028 ms	0.28 μs	0.09%	20 dB

**Table 3 sensors-25-06953-t003:** The measured pinning voltage at the four taps with different V_RES-L_.

		Tap 1	Tap 2	Tap 3	Tap 4
V_RES-L_					
0.5 V		2.22 V	2.22 V	2.24 V	2.33 V
1.5 V		2.22 V	2.22 V	2.23 V	2.32 V
1.7 V		2.25 V	2.26 V	2.24 V	2.32 V

**Table 4 sensors-25-06953-t004:** Measured maximum light intensity at saturation (*E_MAX_*) and gamma of the 4-tap system while all taps are turned OFF and drain turned ON during exposure.

	E_MAX_ [cd/m^2^]	Gamma [-]
Tap 1	752	0.98
Tap 2	1727	0.96
Tap 3	**7408**	**0.94**
Tap 4	4255	0.95

**Table 5 sensors-25-06953-t005:** Luminance required to reach a 1000 LSB signal level at each tap.

		Luminance at 1000 LSB [cd/m^2^]			
The Active Tap		Tap 1	Tap 2	Tap 3	Tap 4
Tap 1		0.3	351	**1504**	94
Tap 2		188	0.3	**992**	701
Tap 3		163	376	0.3	806
Tap 4		175	285	**2280**	0.3

**Table 6 sensors-25-06953-t006:** Measured DR and gamma of the four taps with reset gate low voltage range from 0.5 V to 1.7 V.

		Measured DR or DR Extension [dB]/Gamma [-]		
		V_RES-L_ = 0.5 V	1.5 V	1.7 V
First tap		55.9/1.00	55.9/1.00	**55.9/1.00**
Second tap		19.2/0.98	19.2/0.98	**19.2/0.99**
Third tap		18.6/0.88	18.6/0.88	**18.6/0.89**
Fourth tap		18.7/0.65	19.4/0.83	**19.9/0.87**
Total DR		112.5	113.1	**113.6**

**Table 7 sensors-25-06953-t007:** Charge-splitting HDR with the shortest time window. (Previous setting/ new setting for fourth tap).

		Total Exposure Time	Cycle Exposure Time (τ)	Expected DR Extension

First tap		28 ms	280 μs	-
Second tap		2.8 ms	28 μs	20 dB
Third tap		0.28 ms	2.8 μs	20 dB
Fourth tap		0.028 ms/**2 μs**	280 ns/**20 ns**	20 dB/**42.9 dB**

**Table 8 sensors-25-06953-t008:** Measured DR and gamma of the 4-tap image sensor with different cycle exposure time at the fourth tap.

		Total DR [dB]	Fourth Tap	
$\tau_{4}$			Measured DR/ Ex pected DR [dB]	Gamma [-]
280 ns		113	19.9/20.0	0.87
20 ns		**126**	**32.9/42.9**	**0.71**
Draining		-	37.8	0.94

**Table 9 sensors-25-06953-t009:** Sensor performance summary and comparisons.

	This Work	[17]	[18]	[19]
Technology [nm]	110	45	N/A	45
Pixel count	648 (H) × 480 (V)	3840 (H) × 2160 (V)	1.3 MP	5.7 MP
Pixel pitch [µm]	16.8	2.1	3.0	1.5, 3.0
HDR method	**Charge-splitting**	LOFIC and TCG *	Dual PD and LOFIC & DCG	Dual PD and LOFIC
Single exposure DR	**55.9–126**	110	120	106
Min. SNR at transition [dB]	30 @ 25 °C	25 @ 125 °C	30 @ 100 °C	30 @ 85 °C
Dark noise (random noise)	69 e^−^	622 µV	N/A	1.4 e^−^
FWC [e^−^]	40 k	600 k	N/A	280 k
Cycle exposure time [µs]	0.02–280	-	-	-
Total exposure time [ms]	0.0002–28	10	16	11

* TCG: Triple conversion gain

## Data Availability

The original contributions presented in this study are included in the article. Further inquiries can be directed to the corresponding author.

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
