# Peer review of "Performance Characterization and Tuning of a Charge-Splitting High Dynamic Range 4-Tap CMOS Image Sensor [Author-notes fn1-sensors-25-06953]"

_sensors, 2025, doi:10.3390/s25226953_

Round 1

Reviewer 1 Report

Comments and Suggestions for Authors

Dear Authors,

Thanks for submitting your work to the MDPI sensors. I think the proposed scheme is unique, and enough theory, circuit, operation, and measurement results are well described in this paper. Also, this is one of the good examples of using a ToF image sensor for a normal imaging system. Therefore, it is worth publishing in MDPI sensors. However, some minor modifications/explanations should be added before publication.

(L106-107) This sentence may not be necessary in the technical paper.

(Fig. 1(a)) I suggest adding reset transistors in this layout figure to align with Fig. 1(b).

(Fig. 1(a)) One of the drawbacks of this scheme is FD conversion gain mismatch between 4 taps. Could you add an explanation about it?

(Table 1) I suggest adding a chip micrograph of the fabricated chip.

(Table 1) In my understanding, “Full well capacity” of this table is FDFWC. Is it the average value of 4 taps? Could you add the note to this table?

(Fig. 2) Could you add the readout timing of “RST signal”? Does this sensor operate in differential delta sampling mode (=SIG readout before RST readout)?

(L207-208) I guess you mentioned FD PLS in this sentence. However, I cannot know the location of “A p-n junction between the epi-layer and the FD”. Could you point it out in Fig. 4(b)?

(Fig. 3) In this figure, FDFWC may not be determined by V_RES-L because the potential of V_G-L and “Corridor” is higher than that of V_RES-L.

(Fig. 4(a)) I suggest adding the simulation results of the potential distribution and the charge transfer path.

(Fig. 4(b)) I am confused about this figure. First, why is the depth of n2 deeper than n1? Is n2 used for charge transfer assist? Second, are there n2 and p layers under the position “A”? What is the purpose of them? Third, according to Fig. 4(a), the left edge of the right side n2 is much far from the gate. However, in Fig. 4(b), the edge is the same as the gate.

(L227) I suggest adding some reference papers on “Potential measurements through charge injection” method.

(Fig. 5) I suggest adding pulse timing to this measurement to make clear the method.

(L245) Could you add an explanation about the mechanism of the relationship between the pinning voltage and the reset low voltage?

(L250) Could you add an explanation about the mechanism of this result?

(Fig. 7) I suggest adding pulse timing to this measurement to make clear the method.

(Fig. 7) There are some questions about this result. What is the root cause of the difference between the tap 1 and 2 (Tap 3 and 4)? Why does the linearity fluctuate? Why is the saturation signal level decreased after saturation? Why is the output value so high even with TG=OFF and Drain=ON?

(Fig. 7) Do you have another evaluation result with Drain=OFF during exposure? It is more direct of the actual operation.

(Fig. 8) I suggest adding pulse timing to this measurement to make clear the method.

(Fig. 8(b)) Why is tap 3 not affected by tap 2?

(Fig. 8(c)) What is the root cause of the difference between tap 1 and 2?

(L362-363) -1.8V is much lower than some conventional CIS cases. I suspect it affects the GIDL of the FD node.

(Fig. 11(a)) What does “Draining” mean?

(L410) Before the discussion part, I suggest adding “sample images” captured by this sensor to make clear the effectiveness of this sensor.

Reviewer 2 Report

Comments and Suggestions for Authors

Title: Performance Characterization and Tuning of a Charge-splitting High Dynamic Range 4-Tap CMOS Image Sensor

Line 12: „This is an expanded version of...” Better: This is an extended version of...

Line 38: „... and light emitting diode (LED) flicker, rendering it unsuitable for dynamic scenes.” Here, rendering is not correct used. Rendering meaning represent the process of generation of a digital image, not when you capture a real image.

Line 188: „Under high illumination, this reset transistor acts as an anti-blooming drain...” What is high illumination? Units of measures?

Line 260: „...of the sensor at varying settings using a light source box (Kyoritsu Electric, LB-8623)...” No, Kyoritsu Electric, LB-8623 is not a light source box. It is a Luminance Box, with output between  0.0075 and 18,250 cd/m².

Line 262: „The exposure time was 31.11 ms, and a 50mm / F2.0 VIS-NIR lens (Edmund Optics) was used.” Very good, but the luminance level is missing.

Line 290: Figure 7 - „Light intensity versus pixel value curve for...” I understand, using a.u. arbitrary units. But you have the total control of the measurement chain, you can find the equations from exterior luminance, lens and F numbers, exposure time.

Line 306: „...as it takes the highest light intensity to saturate...” It is not light intensity about. The paper and the research is very hight, to bad to neglect the lighting engineering. Light intensity will produce an incident illuminationg, depending on other parameters.

The research is interesting, with a progressive exposure for each pixel, and not necessarily different exposures for the entire sensor. The work provokes dialogue and even if many details are interpretable, the interdisciplinary area remains a challenge. Lighting engineering is not approached from a metrological point of view, but this aspect may remain to be clarified in the next stages.

Reviewer 3 Report

Comments and Suggestions for Authors

Dear Authors,

Thank you for sharing this interesting and unique pixel design concept and its application to HDR imaging. The paper is well written with excellent English language and logically structured. Beyond the technical value of the work, this paper contributes methodical characterization techniques that junior researchers can learn from and are valuable to publish. I think this paper is worthy of publication in Sensors with some minor revisions and additions.

This paper needs a summary table to compare this work to the state of the art, likely in the discussion section, to place the work in context of HDR imaging.  This is typically done in the imaging field with a comparison table comparing the key performance index (in table 1 of this paper) to the references. This will enable readers to quickly draw conclusions regarding the work.  This pixel is unique and so it is reasonable that the pitch or the read noise will not be as low as some competing sensors and this does not detract from this research or make it any less publishable in Sensors.

Image lag is also not discussed or results presented in this paper which is unusual not to mention in a paper on pixel technology. This is a very unique pixel design concept and achieving zero image lag and complete depletion is challenging in pixel design. If the pixel is lag free then it is an achievement that should be highlighted in the work. Please add comments on, or reference to a paper with the lag measurements, or ideally measurements of image lag to the paper. If there is lag, it’s OK, and does not detract from publication – this is a research project and lag free CIS development is challenging.

Regarding equation 1, on the dynamic range of an image sensor, my understanding is that this is typically understood as 20 log10 (full well capacity/read noise) from e.g. James R. Janesick, Photon transfer : DN --> [lambda], SPIE 2007. Please explain why equation 1 is used.

Regarding Table 1:

  1. Is the FWC quoted from a Photon Transfer Curve? Is it linear FWC, saturation FWC, or ADC-limited FWC?
  2. At what gain is the read noise reported?
  3. Is it possible to share the dark current?

For Figures 7, 8, 9, 10,  and 11, why is light intensity in arbitrary units? What is gained or protected by the authors for using arbitrary units?  It is not as if this is protecting key process information. Responsivity could be calculated by readers from an accurate X-axis scale but the QE is already included in Table 1. It is my opinion that not including units undermines the engineering and scientific soundness of the paper.

I really like the methodical way in which the sensor was characterized and in selecting the tap assignment based on their sensitivity. It appears in the paper that only one chip was measured. Is it known how any other chips perform? Do they have the same relative tap sensitivity offset? I would suspect that each chip would have a unique process and overlay variation leading to different tap relative performance. This would mean that tap tuning would have to be done for each chip separately. If the authors agree, it is worth discussing the practicalities of this.

I think that in the discussion section is the best place to add a summary table comparing the KPI of this work to cited references 13-15 to the paper.

Finally, why do you write this “and could result in dark pixels if the reset transistor activates during exposure.” – why would the reset transistor activate during exposure? Is there a problem with the chip? I agree that reset transistor low voltage increase would decrease the FWC, as you write in the first part of the sentence.
